# The Genotoxicity of Acrylfentanyl, Ocfentanyl and Furanylfentanyl Raises the Concern of Long-Term Consequences

**DOI:** 10.3390/ijms232214406

**Published:** 2022-11-19

**Authors:** Sofia Gasperini, Sabrine Bilel, Veronica Cocchi, Matteo Marti, Monia Lenzi, Patrizia Hrelia

**Affiliations:** 1Department of Pharmacy and Biotechnology, Alma Mater Studiorum University of Bologna, 40126 Bologna, Italy; 2LTTA Center and University Center of Gender Medicine, Department of Translational Medicine, Section of Legal Medicine, University of Ferrara, 44121 Ferrara, Italy; 3Collaborative Center of the National Early Warning System, Department for Anti-Drug Policies, Presidency of the Council of Ministers, 00186 Rome, Italy

**Keywords:** Acrylfentanyl, Ocfentanyl, Furanylfentanyl, Fentanyl, new psychoactive substances, new synthetic opioids, genotoxicity, in vitro mammalian cell micronucleus test, flow cytometry

## Abstract

Three fentanyl analogues Acrylfentanyl, Ocfentanyl and Furanylfentanyl are potent, rapid-acting synthetic analgesics that recently appeared on the illicit market of new psychoactive substances (NPS) under the class of new synthetic opioids (NSO). Pharmacotoxicological data on these three non-pharmaceutical fentanyl analogues are limited and studies on their genotoxicity are not yet available. Therefore, the aim of the present study was to investigate this property. The ability to induce structural and numerical chromosomal aberrations in human lymphoblastoid TK6 cells was evaluated by employing the flow cytometric protocol of the in vitro mammalian cell micronucleus test. Our study demonstrated the non-genotoxicity of Fentanyl, i.e., the pharmaceutical progenitor of the class, while its illicit non-pharmaceutical analogues were found to be genotoxic. In particular, Acrylfentanyl led to a statistically significant increase in the MNi frequency at the highest concentration tested (75 μM), while Ocfentanyl and Furanylfentnyl each did so at both concentrations tested (150, 200 μM and 25, 50 μM, respectively). The study ended by investigating reactive oxygen species (ROS) induction as a possible mechanism linked to the proved genotoxic effect. The results showed a non-statistically significant increase in ROS levels in the cultures treated with all molecules under study. Overall, the proved genotoxicity raises concern about the possibility of serious long-term consequences.

## 1. Introduction

Fentanyl is a potent, rapid-acting synthetic analgesic, first developed in 1960 by Dr. Paul Janssen and the Janssen Company [1]. Due to its high potency, efficacy and rapid action, Fentanyl has become one of the world’s most important opioids and is frequently used to manage acute and chronic pain conditions. Indeed, Fentanyl is about 50–100 times more potent than the well-known natural opioid morphine [2]. Similarly to other opioids, Fentanyl acts as a stimulant of the μ-opioid receptors and typically induces central nervous system effects including euphoria, sedation, fatigue, nausea, vomiting, dizziness, respiratory depression, bradycardia and anesthesia at high dosage [3]. Following Fentanyl’s discovery, several structural analogues have been designed by various pharmaceutical companies to enhance the pharmacological action of these drugs [3]. A few years after Fentanyl and its analogues’ approval as medicines, clinical misuse and illicit use were reported [2]. Deaths from Fentanyl overdose have been due to inappropriate prescriptions by clinicians, patients’ misuse and increased illicit use and abuse [3]. More recently, Fentanyl appeared on the illicit market of new psychoactive substances (NPS) under the class of novel synthetic opioids (NSO) [4]. Fentanyl and its non-pharmaceutical analogues are easily produced in clandestine laboratories and are sold in different forms such as powder or tablets to be injected, ingested, snorted or smoked, often sold as an alternative or adulterant to heroin [4,5]. To date, about one thousand Fentanyl analogues have been reported to the European Monitoring Centre for Drugs and Drugs Abuse (EMCDDA) [6]. In 2017, 940 seizures of Fentanyl derivatives were reported for a total of 14,3 tons in 13 countries of the European Union [5].

Fentanyl analogues contribute to the current epidemic of opioid overdose deaths in many countries [4,5,6]. Overdose death related to Fentanyl analogues is mainly due to the unpredictability of their potency. Moreover, in some cases of Fentanyl-related deaths involving Acrylfentanyl, Ocfentanyl and Furanylfentanyl (Figure 1), no heroin was detected [7].

Acrylfentanyl, N-phenyl-N-[1-(2-phenethyl)piperidin-4-yl]prop-2-enamide, was first developed in 1981 as an unsaturated fentanyl analogue [8]. It is an acrylamide derivative of 4-anilinopiperidine. In the last few years, it has appeared on the Surface Web as a research chemical. In 2016, the EMCDDA reported a total of 130 deaths caused by Acrylfentanyl in the United States and Europe [9].

Ocfentanyl, N-(2-fluorophenyl)-2-methoxy-N-[1-(2-phenylethyl)piperidin-4-yl]acetamide, is a fentanyl analogue which has an extra fluorine atom on the o-position of the aniline moiety and a methoxy group instead of a methyl group. It had been studied clinically for its analgesic activity during the early 1990s and was found to have a relative potency 200 times greater than that of morphine and 2.5 times more potent than Fentanyl, with lower cardio-respiratory toxicity. However, it has never been approved for medical use [10]. Ocfentanyl was among the 14 Fentanyl derivatives that were reported in Europe’s illicit market between 2012 and 2016. It has been associated with cases of intoxication and death in Belgium and Switzerland [11].

Furanylfentanyl, N-phenyl-N-[1-(2-phenylethyl)piperidin-4-yl]furan-2-carboxamide, is another Fentanyl analogue, first described in 1958, which differs from Fentanyl by the presence of a furanyl ring in place of the methyl group adjacent to the carbonyl bridge. Furanylfentanyl showed a seven-fold higher potency compared with fentanyl, but it has not been approved for any medical use [10]. Since 2015, Furanylfentanyl has been identified in the United States and in 16 member states of the European Union. In 2016, a total of 494 forensic cases regarding Furanylfentanyl, including 128 confirmed fatalities, were reported to the DEA. In 2017, the EMCDDA reported 10 acute intoxications and 19 deaths associated with Furanylfentanyl in Europe and Norway [12].

In general, pharmacotoxicological data on these compounds is limited, but a recent study on the pharmacodynamics of Fentanyl and its three analogues revealed that Acrylfentanyl, Ocfentanyl and Furanylfentanyl behave similarly to Fentanyl as μ opioid agonists. In particular, Acrylfentanyl, Ocfentanyl and Furanylfentanyl acted as partial agonists on μ opioid receptors in vitro. Moreover, in the BRET G-protein assay, Fentanyl, Acrylfentanyl and Ocfentanyl behaved as partial agonists for the β-arrestin 2 pathway, whereas Furanylfentanyl did not promote β-arrestin 2 recruitment. In the same study, Fentanyl, Acrylfentanyl and Ocfentanyl increased mechanical and thermal analgesia and impaired motor and cardiorespiratory parameters in vivo, while Furanylfentanyl showed lower potency for cardiorespiratory and motor effects [13].

Moreover, another study demonstrated that larvae fish exposed for 24 h to fentanyl at different concentrations of 1, 10, 50, 100 μM survived, although they manifested several morphological malformations [14].

Regarding the genotoxicological hazards of fentanyls, to our knowledge, only one study has been published concerning Fentanyl and two of its pharmaceutical analogues (Alfentantil and Remifentanil) [15], while studies on the genotoxicity of non-pharmaceutical fentanyl analogues, including Acrylfentanyl, Ocfentanyl and Furanylfentanyl, are not yet available.

Therefore, in order to fill this gap, the aim of the present study was to evaluate the genotoxicity of these three fentanyl analogues and their progenitor Fentanyl in human lymphoblastoid TK6 cells, particularly in terms of the ability to induce structural and numerical chromosomal aberrations. For this purpose, we evaluated the frequency of micronuclei (MNi) by employing a flow cytometric (FCM) protocol developed in our laboratory and published [16,17] as the in vitro mammalian cell micronucleus test (MNvit), correspondent to OECD guideline n°487. We selected the concentrations to be tested based on cytotoxicity thresholds established by OECD and on apoptosis induction, checked as an alternative cell death mechanism to necrosis [18].

## 2. Results

### 2.1. Cytotoxicity Evaluation

According to OECD guideline n°487, the highest concentration to be tested for MNi frequency evaluation should not produce cytotoxicity levels greater than 55 ± 5%, and consequently it is recommended to proceed only if the treated populations display cell viability and cell proliferation of at least 45 ± 5% when compared with concurrent negative control cultures [18]. Figure 2 shows the compliance with the OECD guideline threshold (represented by the red line) of the cytotoxicity induced by all fentanyls at all the concentrations tested.

Meanwhile, correct cell replication was also checked by calculating the relative population doubling (RPD), whose formula is reported in Materials and Methods Section 4.4.1 [18].

RPDs were above the OECD guideline threshold (equal to 45 ± 5%) for Fentanyl and Ocfentanyl at all the concentrations tested, for Acrylfentanyl up to 100 µM and for Furanylfentanyl up to 50 µM (Table 1).

### 2.2. Apoptosis Evaluation

Apoptosis induction was measured to analyze an alternative cell death mechanism to necrosis and to check if apoptosis levels in treated cultures were comparable or up to twice the values measured in the concurrent negative cultures. In particular, the two or three highest concentrations that complied with the OECD cytotoxicity thresholds were evaluated for each substance: 100, 150 and 200 µM for Fentanyl, 50, 75 and 100 µM for Acrylfentanyl, 100, 150 and 200 µM for Ocfentanyl and 25 and 50 µM for Furanylfentanyl. Etoposide (ETP) 5 µg/mL was used as positive control.

Apoptosis never doubled compared to that measured in the concurrent negative control for all fentanyls tested, except for Acrylfentanyl 100 µM (2.5 vs. 1.0, but not statistically significant) (Figure 3).

### 2.3. MNi Frequency Evaluation

Based on the results obtained from viability, RPD and apoptosis, two concentrations to be tested for MNi frequency evaluation were selected for each compound: 150 and 200 µM for Fentanyl, 75 and 100 µM for Acrylfentanyl, 150 and 200 µM for Ocfentanyl and 25 and 50 µM for Furanylfentanyl.

To evaluate the potential genotoxic effect associated with these fentanyls, the number of MNi was measured in untreated negative control cultures, fentanyls-treated cultures and positive control-treated cultures, i.e., treated with Mytomicin C (MMC) or Vinblastine (VINB).

Fentanyl was the only molecule that did not cause a statistically significant increase in the MNi frequency fold increase at any of the concentrations tested. On the contrary, Acrylfentanyl determined a statistically significant increase only at the highest concentration tested, while Ocfentanyl and Furanylfentnyl did so at both the concentrations tested (Figure 4A,D,G,J).

In Figure 3, representative FCM dot plots are reported to support the results obtained. In particular, they show how the numbers of MNi were much higher in Acrylfentanyl, Ocfentanyl or Furanylfentanyl-treated cultures (Figure 4F,I,L) than in the concurrent negative control cultures (Figure 4E,H,K), while in Fentanyl-treated cultures (Figure 4C) they were comparable to the concurrent negative control cultures (Figure 4B).

### 2.4. ROS Evaluation

In order to identify a possible mechanism linked to the proved genotoxic effect, TK6 cells were treated with all fentanyls under study for 1 h, and then possible ROS induction was measured.

For each substance, the highest concentration tested for the MN frequency evaluation was selected for ROS evaluation: 200 µM for Fentanyl, 100 µM for Acrylfentanyl, 200 µM for Ocfentanyl and 50 µM for Furanylfentanyl. For all fentanyls-treated cultures, the results did not show a statistically significant increase of the mean fluorescence intensity compared with the concurrent negative controls (Figure 5).

## 3. Discussion

The protection of public health necessarily includes the risk assessment of any substance to which humans are potentially exposed. The risk assessment, in turn, necessarily includes the evaluation of genotoxicity and carcinogenicity. While this concept is well-known and strictly regulated for pharmaceutical drugs, cosmetics, pesticides and food additives, it has not been so frequently studied for drugs of abuse. In general, and rightly so, careful attention is given to the management of the intoxicated patient and to understanding and containing the acute toxic effects. However, we must also pay close attention to the potential long-term effects of genotoxicity, which can induce permanent transmissible changes in the amount or structure of the genetic material, and can occur as early as a single exposure and with increased likelihood following repeated exposure [19].

The available toxicological data are limited regarding the non-pharmaceutical illicit fentanyl analogues Acrylfentanyl, Ocfentanyl and Furanylfentanyl, and genotoxicological data are totally absent. Therefore, in this work we investigated their genotoxic potential and that of their progenitor Fentanyl.

To assess genotoxicity, first we defined the concentrations to be tested of all fentanyls under study, based on absent or poor cell death and replicative capacity [18]. OECD guideline n°487 established a cytotoxicity threshold equal to 55 ± 5%, which consequently corresponds to a cell viability and proliferation of at least 45 ± 5% when compared to concurrent negative control cultures [18].

For this purpose, we applied propidium iodide (PI) staining to determine the percentage of viable cells in untreated cultures and cultures treated with Fentanyl, Acrylfentanyl, Ocfentanyl or Furanylfentanyl scalar concentrations from 0 to 200 µM. The results obtained showed that all fentanyls complied with the OECD threshold at all the concentrations tested.

In parallel, by the same assay we measured the number of cells at time zero (time of cellular seeding) and at the end of the treatment time, to check correct cell proliferation. The threshold was met by Fentanyl and Ocfentanyl at all the concentrations tested, while by Acrylfentanyl up to 100 µM and by Furanylfentanyl up to 50 µM.

PI staining makes it possible to discriminate between viable and necrotic cells, but it does not highlight apoptotic cells. The distinction is based solely on the difference in membrane integrity and consequent permeability to PI. For this reason, apoptotic cells, characterized by a still intact membrane, could be misinterpreted by the instrument as viable cells. Therefore, we proceeded with a more specific test to highlight this alternative death mechanism. We performed a double staining 7-AAD/Annexin V-PE assay to measure the percentage of apoptotic cells in untreated cultures as well as in those treated with Fentanyl 100, 150 and 200 µM, Acrylfentanyl 50, 75 and 100 µM, Ocfentanyl 100, 150 and 200 µM and Furanylfentanyl 25 and 50 µM, corresponding to the two or three highest concentrations selected from those meeting the cytotoxicity thresholds. The results obtained demonstrated that apoptosis never as much as doubled compared to that measured in the concurrent negative control, for all compounds except for Acrylfentanyl 100 µM but without statistical significance. This outcome is particularly important from a genotoxicological point of view. The process of apoptosis could be triggered as a consequence of unrepaired genetic damage, induced by genotoxic agents. However, resistance to apoptosis can result in the inability of cells through this selective cell elimination to counteract the transmission of genetic damage from mother cell to daughter cells [20,21].

Having identified suitable concentrations, we evaluated genotoxicity in terms of induction of chromosomal aberrations, selecting the MNvit test for this purpose. The *Overview of the set of OECD genetic toxicology test guidelines* lists the genotoxicity tests validated by OECD [22]. Among them, those especially recommended by most regulatory agencies and international authorities and most frequently used by the scientific community are the bacterial reverse mutation test (Ames test), corresponding to OECD guideline n°471, and the MNvit, corresponding to OECD guideline n°487. The former identifies chemicals that induce gene mutations, the latter allows highlighting of structural chromosomal aberrations induced by clastogenic agents and numerical chromosomal aberrations induced by aneugenic agents. The MNvit test therefore owes its success to scientific validity, versatility and efficacy. However, the classic method of analysis by optical microscopy is affected by certain critical issues, such as the subjectivity of interpretation, long sample preparation and analysis times, and the number of events examined which is extremely low for the purpose of robust statistical analysis. To overcome these limits, we developed and published in 2018 an innovative protocol to score MNi by flow cytometry. Since then, our protocol has been successfully used to prove the genotoxicity of different NPS [23,24,25,26] and the lack of genotoxic capacity of other types of substances [17,27].

In the present study our protocol permitted us to demonstrate the inability of Fentanyl to increase statistically significantly the frequency of MNi.

To our knowledge, only one study is available in the literature regarding the genotoxicological evaluation of this drug in vitro and in vivo, reporting a positive outcome only in the in vitro L5178Y *tk*^+/−^ mouse lymphoma assay (MLA) in the presence of rat liver metabolic activation (S9), and negative outcomes in other genotoxicity tests such as the Ames test, the unscheduled DNA synthesis (UDS) assay on primary rat hepatocytes and the in vitro chromosomal aberration (CA) assay on human lymphocytes and CHO cells. Moreover, the same authors also reported that Fentanyl did not cause a positive response in the in vitro carcinogenicity BALB/c-3T3 transformation test. These results substantially agree with our results, except for the MLA findings.

Regarding the three non-pharmaceutical illicit fentanyl analogues tested in the present work, contrary to their progenitor, our results demonstrated the genotoxic ability of Ocfentanyl and Furanylfentanyl at both the concentrations tested, and that of Acrylfentanyl at the highest concentration tested.

These results may seem to conflict partially with what was demonstrated regarding two other pharmaceutical Fentanyl analogues, Alfentanil and Remifentanil [15]. In particular, Alfentanil determined a non-genotoxic response in the Ames test and in an in vivo MN test [15]. The pharmaceutical analogue Remifentanil was revealed to be non-genotoxic according to the Ames test, an in vitro CA assay in CHO cells, an in vivo MN assay in rat erythrocytes and in vivo/in vitro UDS assay in rat hepatocytes. However, similarly to Fentanyl, Remifentanil also produced a weak genotoxic response only in the in vitro L5178Y *tk*^+/−^ MLA in the presence of rat liver S9 metabolic activation [15]. All these outcomes are summarized in Table 2.

These contrasting results for Fentanyl analogues are an indication of molecules characterized by structural differences.

In light of this, it would be interesting to investigate the relationship between fentanyls’ structures and their genotoxic potential. However, this is not an easy task considering the scarcity of data on this specific aspect of fentanyls.

More generally, regarding the prediction of chemicals’ genotoxicity, in silico studies reported aniline and substituted anilines present in all fentanyls, among the representative structural alerts which are present in MN-positive chemicals [28,29]. Furthermore, another innovative in silico study predicting in vitro MN induction individuated tertiary aliphatic and aromatic amines, other functional groups present in all fentanyls, as genotoxic-related structural alerts. Moreover, the same study revealed among these structures α,β-unsaturated compounds, including the acrylamide group, present in acrylfentanyl [29]. Given the different genotoxicological outcomes obtained in our experiments for the four fentanyls, it could be hypothesized that they can be attributed to the chemical reactivity and/or steric hindrance of the different functional groups added to the basic structure of fentanyl. This hypothesis should be demonstrated through specific studies of the structure–activity relationship, but it reminds us of a well-known pharmacological concept, i.e., a small structural change can lead to very different biological effects or even none at all. This fact highlights the importance of testing the genotoxic potential not only of a few representative molecules of a class, but of every single molecule belonging to it, including even slight differences in structure or conformation, as previously demonstrated for other NPS classes [24,26]. Indeed, there is the possibility of different biological effects within the same structural class, as already observed, for example, in our previous study on synthetic cathinones [25].

The genotoxic effect observed lead us to investigate the possible mechanism underlying genotoxicity. For this purpose, the study continued by analyzing ROS induction. It is indeed well recognized how ROS, such as singlet oxygen ^1^O_2_, superoxide radicals ·O_2_^−^, hydrogen peroxide H_2_O_2_ and hydroxyl radical ·OH, are involved in genetic damage.

After 1 h treatment with the highest doses of the fentanyls tested for MNi frequency evaluation, intracellular ROS levels were measured by means of DCF assay. Our results revealed no statistically significant increase in ROS levels for any of the fentanyls analyzed and, albeit preliminary, suggest the involvement of other mechanisms underlying the proven genetic damage.

## 4. Materials and Methods

### 4.1. Reagents

2′,7′-dichlorodihydrofluorescin diacetate (DCFH_2_-DA), Ethylenediaminetetraacetic acid (EDTA), Etoposide (ETP), fetal bovine serum (FBS), L-Glutamine (L-GLU), Hydrogen peroxide (H_2_O_2_), Mitomycin C (MMC), Nonidet, Penicillin-Streptomycin solution (PS), Phosphate-Buffered Saline (PBS), Potassium Chloride, Potassium Dihydrogen Phosphate, Roswell Park Memorial Institute (RPMI) 1640 medium, Sodium Chloride, Sodium Hydrogen Phosphate, Vinblastine (VINB), water bpc grade (all purchased from Merck, Darmstadt, Germany), Guava Nexin reagent (containing 7-aminoactinomycin (7-AAD) and Annexin-V-PE), Guava ViaCount reagent (containing PI) (all purchased from Luminex Corporation, Austin, Texas, USA), RNase A, SYTOX Green (purchased from Thermo Fisher Scientific, Waltham, MA, USA).

### 4.2. Fentanyls

Fentanyl, Acrylfentanyl, Ocfentanyl and Furanylfentanyl were purchased from LGC Standards (LGC Standards S.r.L., Sesto San Giovanni, Milan, Italy) and www.chemicalservices.net (accessed on 5 October 2022). The test compounds were dissolved in absolute ethanol up to 20 mM stock solution and stored at −20 °C. Absolute ethanol concentration was maintained in the range 0–1% in all experimental conditions, to avoid potential solvent toxicity.

### 4.3. Cell Culture and Treatments

TK6 Human lymphoblastoid cells were purchased from Merck (Darmstadt, Germany) and were grown at 37 °C and 5% CO_2_ in RPMI-1640 supplemented with 10% FBS, 1% L-GLU and 1% PS. To maintain exponential growth and considering that the time required to complete the cell cycle is 13–14 h, the cultures were divided every three days in fresh medium and the cell density did not exceed the critical value of 9 × 10^5^ cells/mL.

In all the experiments, aliquots of 2.5 × 10^5^ of TK6 cells were treated with increasing concentrations of Fentanyl, Acrylfentanyl, Ocfentanyl and Furanylfentanyl included in the range 0–200 µM and incubated for 26 h, corresponding to 1.5–2 replication cycles of the TK6 cells.

Cytotoxicity, apoptosis and MNi frequency evaluation were measured at the end of the 26 h treatment time.

### 4.4. Flow Cytometry

All FCM analysis reported below were performed using a Guava easyCyte 5HT flow cytometer equipped with a class IIIb laser operating at 488 nm (Luminex Corporation, Austin, TX, USA).

#### 4.4.1. Cytotoxicity Analysis

Cytotoxicity assay was performed as previously described by Lenzi et al. [30,31]. Briefly, cells were stained with PI and 1000 cells per sample were analyzed.

The viability percentage recorded in the treated cultures was normalized with that recorded in the concurrent negative control cultures, considered equal to 100%. These results made it possible to confirm that the cell viability percentage respected the OECD threshold (equal to 45 ± 5%) under all experimental conditions [18].

In parallel, always using PI reagent, the number of cells seeded at time zero and that measured at the end of the treatment time were evaluated, to check the correct replication in the negative control cultures and to compare it using RPD with that measured in the treated cultures. Population doubling (PD) and RPD were calculated with the following formulas:$$\mathrm{PD}=log\left(\frac{post-treatment\;cell\;number}{initial\;cell\;number}\right)\;÷log\;2$$
$$\mathrm{RPD}=\frac{PD\;in\;treated\;cultures}{PD\;in\;control\;cutures}\times 100$$

Similarly to cytotoxicity, cytostasis was checked in order to verify that cell proliferation respected the threshold established by the OECD guideline (equal to 45 ± 5%) [18].

#### 4.4.2. Apoptosis Analysis

The percentage of apoptotic cells was evaluated according to the procedure used by Lenzi et al. [30,31].

Briefly, the percentage of apoptotic cells was assessed by means of a double-staining protocol with 7-AAD and Annexin-V-PE, analyzing 2000 cells per sample.

The percentage of apoptotic cells recorded in the treated cultures was normalized with that recorded in the concurrent negative cultures, considered equal to 1, and expressed as the apoptotic fold increase. These results were checked to ensure that the apoptosis induction was similar to at most double that recorded in the concurrent negative cultures. A concentration of 5 µg/mL of ETP was used as positive control.

#### 4.4.3. MNi Frequency Evaluation

The analysis of MNi frequency was performed using an automated protocol developed in our laboratory [16] and already successfully employed [17,23,24,25,26,27].

Briefly, at the end of the treatment time cells were collected, lysed and stained with SYTOX Green. The discrimination between nuclei and MNi was performed based on size difference analyzed by forward scatter (FSC) and the intensity of green fluorescence.

The MNi frequency, calculated as the number of MNi per 10,000 nuclei deriving from living and proliferating cells for every sample, and recorded in treated cultures at all the concentrations tested, was normalized with those recorded in the concurrent negative control cultures.

We used the clastogen Mitomycin C (MMC) and the aneugen Vinblastine (VINB) as positive controls [18].

#### 4.4.4. ROS Evaluation

The analysis of ROS levels was performed using the 2′,7′-dichlorofluorescin (DCF) assay as previously described by Cocchi et al. [24,27].

Briefly, at the end of the treatment time (1 h) 2 × 10^5^ cells were stained at 37 °C in the dark for 20 min with 2′,7′-dichlorodihydrofluorescin diacetate (DCFH_2_-DA). A total of 5000 events, derived from viable cells, were acquired and analyzed by Guava Incyte software.

The mean fluorescence intensity values of DCF (which is formed in cells in presence of ROS) recorded in the treated cultures were normalized with that recorded in the untreated control culture, accounted equal to 1 and expressed as ROS fold increase.

We used H_2_O_2_ 100 µM as positive control.

### 4.5. Statistical Analysis

Each concentration of fentanyls was tested in triplicate at all the experimental conditions. All analyses were repeated three times. Viability percentage, RPD, apoptosis fold increase and MNi frequency fold increase were expressed as mean ± SEM. For all experimental conditions, more than three groups of matched data were compared, so statistical significance was analyzed by one-way repeated measures ANOVA, followed by Dunnett or Bonferroni post-testing to compare all treated groups with the control. We considered the difference between means as statistically significant if *p* value < 0.05.

We used Prism Software 4 to carry out the calculations.

## 5. Conclusions

Our study demonstrated for the first time that Acrylfentanyl, Ocfentanyl and Furanylfentanyl are genotoxic, in terms of their ability to induce chromosomal aberrations. A completely exhaustive evaluation of their genotoxic capacity could suggest proceeding by analyzing their ability to induce point mutations, e.g., by means of the Ames test. However, although this could provide additional genotoxicological information, the positive outcome of the MNvit analysis has already highlighted the important aspect of the research, i.e., it has identified an additional alarming toxicological concern related to these three fentanyl derivatives.

Indeed, the demonstrated genotoxic effect advocates raising awareness about the possibility of serious long-term consequences, given the well-known key role played by genotoxicity in the development of numerous neuro- and chronic-degenerative diseases [20,32].

## Figures and Tables

**Figure 1 ijms-23-14406-f001:**
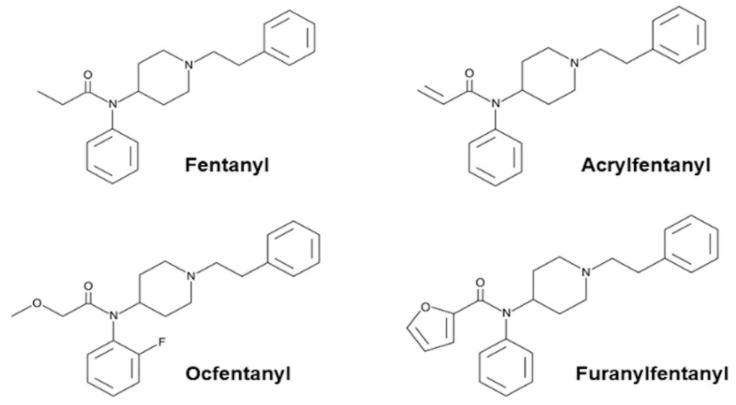
Molecular structures of Fenatnyl, Acrylfentanyl, Ocfentanyl and Furanylfentanyl.

**Figure 2 ijms-23-14406-f002:**
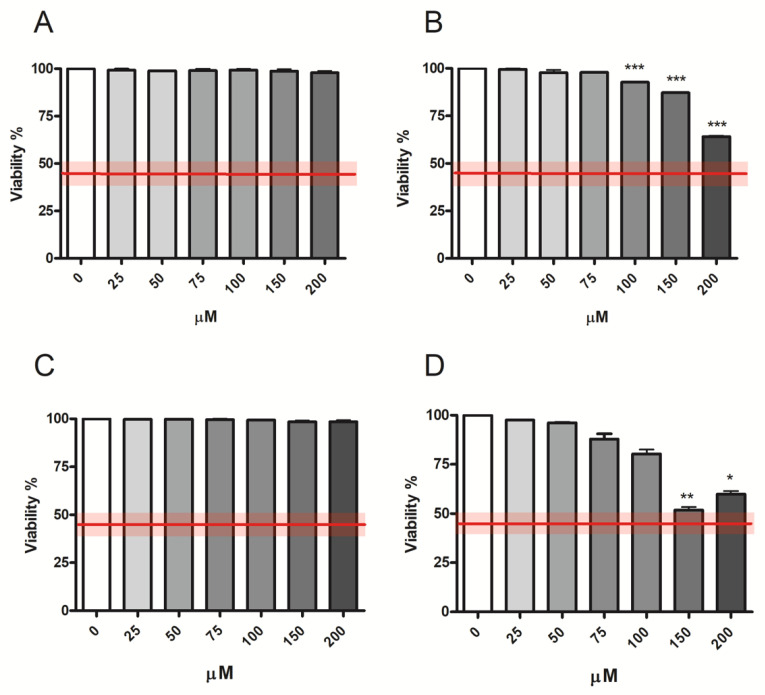
Cell viability of TK6 cells after 26 h treatment with (**A**) Fentanyl, (**B**) Acrylfentanyl, (**C**) Ocfentanyl or (**D**) Furanylfentanylat the indicated concentrations compared to the concurrent negative controls [0 µM]. Each bar represents the mean ± SEM of three independent experiments. Data were analyzed using one-way repeated measures ANOVA followed by Dunnet as post-test. * *p* < 0.05 vs. [0 μM]; ** *p* < 0.01 vs. [0 µM]; *** *p* < 0.001 vs. [0 µM]. The red line represents the OECD threshold for viability (45 ± 5%).

**Figure 3 ijms-23-14406-f003:**
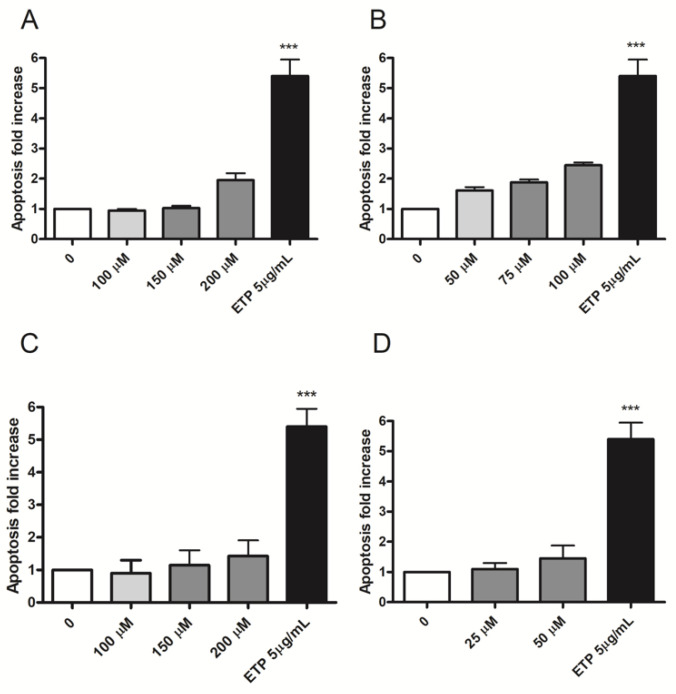
Apoptosis fold increase in TK6 cells after 26 h treatment with (**A**) Fentanyl, (**B**) Acrylfentanyl, (**C**) Ocfentanyl or (**D**) Furanylfentanyl or the positive control ETP at the indicated concentrations, compared to the concurrent negative control [0 µM]. Each bar represents the mean ± SEM of three independent experiments. Data were analyzed using one-way repeated measures ANOVA followed by Bonferroni as post-test. *** *p* < 0.001 vs. [0 µM].

**Figure 4 ijms-23-14406-f004:**
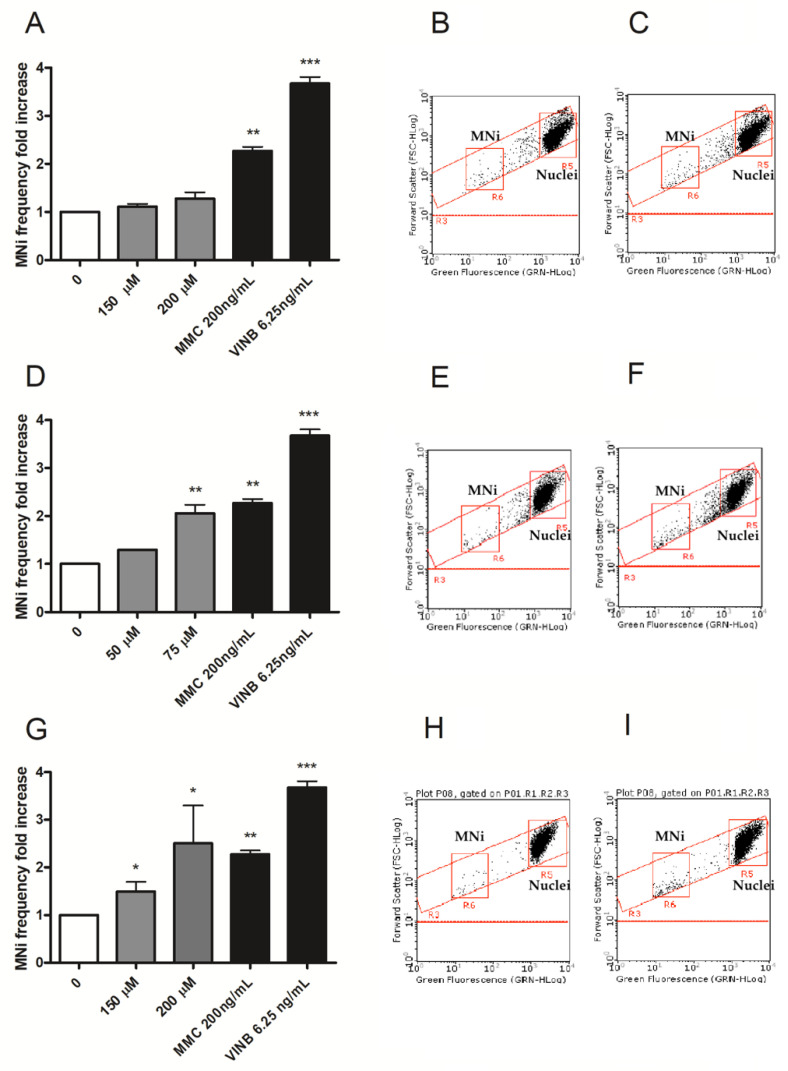
MNi fold increase in TK6 cells after 26 h treatment with (**A**) Fentanyl, (**D**) Acrylfentanyl, (**G**) Ocfentanyl or (**J**) Furanylfentanyl or positive controls (MMC, VINB) at the indicated concentrations compared to the concurrent negative control [0 µM]. Each bar represents the mean ± SEM of three independent experiments. Data were analyzed using one-way repeated measures ANOVA followed by Bonferroni as post-test. * *p* < 0.05 vs. [0 µM]; ** *p* < 0.01 vs. [0 µM]; *** *p* < 0.001 vs. [0 µM]. Dot plots obtained by FCM in (**B**,**E**,**H**,**K**) negative controls, (**C**) Fentanyl 200 µM, (**F**) Acrylfentanyl 75 µM, (**I**) Ocfentanyl 200 µM and (**L**) Furanylfentanyl 50 µM. “MNi” and “Nuclei” indicate the regions of the dot plot where MNi and nuclei can be found.

**Figure 5 ijms-23-14406-f005:**
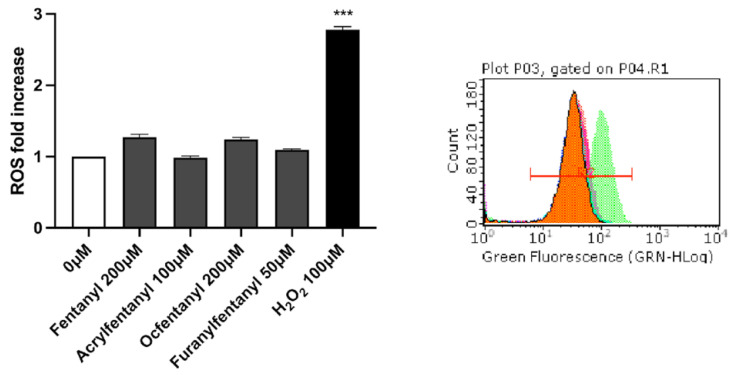
ROS fold increase in TK6 cells after 1 h treatment with Fentanyl, Acrylfentanyl, Ocfentanyl and Furanylfentanyl or positive control [H_2_O_2_] at the indicated concentrations, compared to the concurrent negative control [0 μM]. Each bar represents the mean ± SEM of three independent experiments. Data were analyzed using repeated ANOVA followed by Dunnet as post-test. *** *p* < 0.001 vs. [0 mg/mL]. Histograms obtained by FCM for the negative control (orange), H_2_O_2_ (green) and fentanyls (other colors).

**Table 1 ijms-23-14406-t001:** RPD of TK6 cells after 26 h treatment with Fentanyl, Acrylfentanyl, Ocfentanyl or Furanylfentanyl at the indicated concentrations compared to the concurrent negative control [0 µM]. Each value represents the mean ± SEM of three independent experiments. Data were analyzed using one-way repeated measures ANOVA followed by Dunnet as post-test. * *p* < 0.05 vs. [0 µM]; *** *p* < 0.001 vs. [0 µM]. The red lines separate the RPDs complying or not with the OECD threshold for cell replication (45 ± 5%).

Concentration	RPD Fentanyl	RPD Acrylfentanyl	RPD Ocfentanyl	RPD Furanylfentanyl
0 µM	100.0%	100.0%	100.0%	100.0%
25 µM	95.2 ± 1.1%	96.8 ± 0.3%	99.7 ± 0.2%	98.8 ± 1.3%
50 µM	89.9 ± 1.5% *	91.8 ± 2.3% *	97.1 ± 1.6%	78.4 ± 1.9% ***
75 µM	90.0 ± 2.6%	78.75 ± 2.8% ***	95.5 ± 2.7%	38.3 ± 1.6%
100 µM	91.2 ± 2.9% *	64.1 ± 1.2% ***	94.4 ± 4.1%	20.5 ± 0.5% ***
150 µM	85.5 ± 1.6% ***	40.9 ± 1.1% ***	97.2 ± 1.7%	0.0% ***
200 µM	80.6 ± 1.3% ***	1.5 ± 2.4% ***	93.1 ± 2.4% *	0.0% ***

**Table 2 ijms-23-14406-t002:** Available genotoxicological studies on fentanyls.

Fentanyls	Genotoxicity Tests	Reference
**Fentanyl**	Negative in: ▪ Ames test (on *S. typhimurium* and *E. coli*) ▪ in vivo/in vitro UDS assay (in primary rat hepatocytes) ▪ in vitro CA assay (in human lymphocytes and Chinese hamster ovary (CHO) cells) Positive in: ▪ MLA, only in the presence of S9	[15]
**Alfentanyl**	Negative in: ▪ Ames test (on *S. typhimurium*) ▪ in vivo MN test (in female rats)	[15]
**Remifentanil**	Negative in: ▪ Ames test (on *S. typhimurium*) ▪ in vivo/in vitro UDS assay (in primary rat hepatocytes) ▪ in vitro CA assay (in CHO cells) ▪ in vivo MN test (in rat erythrocytes) Positive in: ▪ MLA, only in the presence of S9	[15]

## Data Availability

Not applicable.

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
