# Peer review of "The Genotoxicity of Acrylfentanyl, Ocfentanyl and Furanylfentanyl Raises the Concern of Long-Term Consequences"

_ijms, 2022, doi:10.3390/ijms232214406_

Round 1
Reviewer 1 Report
The authors revealed three fentanyl analogues Acrylfentanyl, Ocfentanyl and Furanylfentanyl genotoxicity, and the results showed that all three compounds tested are genotoxic. The research is interesting and important. However, some questions might be answered before it can be accepted.
1. Why not use Ames test as one of the genotoxicity?
2. The structure and genotoxicity of three drugs should be discussed deeply.
3. A compared table about the genotoxicity as well as references including this kind of drugs should be added.
Author Response
We thank the reviewer for the helpful suggestions to improve our work. The required information and changes are listed point by point and highlighted in red in the manuscript for faster viewing.
- Why not use Ames test as one of the genotoxicity?
We thank the reviewer for the rightful observation. We have added a more in-depth comment about this aspect in the discussion section.
- The structure and genotoxicity of three drugs should be discussed deeply.
As suggested, we have included a comment on this aspect in the discussion section.
- A compared table about the genotoxicity as well as references including this kind of drugs should be added.
We thank the reviewer for its suggestion. We added the suggested table summarizing the limited available genotoxicological data on fentanyl and its analogues in the discussion section. Unfortunately, only one study on Remifentanil’s genotoxicity is available.

Reviewer 2 Report
The manuscript abstract must be adequate for the journal's standards.
In addition, authors need to provide the most relevant results of the manuscript in the abstract.
The results of the manuscript have some scientific relevance, however, for publication in this journal, authors need to present more results to improve the overall scientific quality of the manuscript.
The discussions presented are somewhat superficial, due to the lack of more results to be analyzed and discussed by the authors.
This makes the manuscript very superficial and without great answers when the investigation carried out by the authors.
The manuscript has no conclusions. Did the authors adequately review the manuscript structure before submission?
Author Response
We thank the reviewer for the helpful suggestions to improve our work. The required information and changes are listed point by point and highlighted in red in the manuscript for faster viewing.
The manuscript abstract must be adequate for the journal's standards.
We thank the reviewer for the rightful observation. We corrected the abstract according to the journal's standards.
In addition, authors need to provide the most relevant results of the manuscript in the abstract.
As suggested, we implemented the abstract by adding the most relevant results.
The results of the manuscript have some scientific relevance, however, for publication in this journal, authors need to present more results to improve the overall scientific quality of the manuscript.
As suggested by the reviewer, we performed additional experiments on the test chemicals to investigate a possible mechanism at the basis of the proved genotoxic effect. In particular, since the induction of reactive oxygen species (ROS) is one of the possible modes of action of genotoxic agents we decided to evaluate this endpoint.
The discussions presented are somewhat superficial, due to the lack of more results to be analyzed and discussed by the authors.
This makes the manuscript very superficial and without great answers when the investigation carried out by the authors.
As suggested by the author, we implemented the discussion section by commenting the new results presented on ROS induction, the choice of the MN test, the relationship between the structure of the molecules under study and genotoxicity and the genotoxicological data available of Fentanyl and other pharmaceuticals analogues
The manuscript has no conclusions. Did the authors adequately review the manuscript structure before submission?
As suggested, we have now added conclusions section after the discussion section.

Round 2
Reviewer 1 Report
It's OK.
Reviewer 2 Report
After the modifications carried out, the manuscript appears to be suitable for publication.